# MarioNette: Self-Supervised Sprite Learning

**Dmitriy Smirnov**
MIT

**Michaël Gharbi**
Adobe Research

**Matthew Fisher**
Adobe Research

**Vitor Guizilini**
Toyota Research Institute

**Alexei A. Efros**
UC Berkeley

**Justin Solomon**
MIT

## Abstract

Artists and video game designers often construct 2D animations using libraries of sprites—textured patches of objects and characters. We propose a deep learning approach that decomposes sprite-based video animations into a disentangled representation of recurring graphic elements in a self-supervised manner. By jointly learning a dictionary of possibly transparent patches and training a network that places them onto a canvas, we deconstruct sprite-based content into a sparse, consistent, and explicit representation that can be easily used in downstream tasks, like editing or analysis. Our framework offers a promising approach for discovering recurring visual patterns in image collections without supervision.

Since the early days of machine learning, the accepted unit of image synthesis has been the *pixel*. But while the pixel grid is a natural representation for display hardware and convolutional generators, it does not easily permit high-level reasoning and editing.

In this paper, we take inspiration from animation to consider an atomic unit that is richer and easier to edit than the pixel: the *sprite*. In sprite-based animation, a popular early technique for drawing cartoons and rendering video games, an artist draws a collection of patches—a *sprite sheet*—consisting of texture swatches, characters in various poses, static objects, and so on. Then, each frame is assembled by compositing a subset of the patches onto a canvas. By reusing the sprite sheet, authoring new content requires minimal effort and can even be automated procedurally.

Our goal is to invert this process, simultaneously tackling unsupervised instance segmentation and dictionary learning. Given an image dataset, e.g., frames from a sprite-based video game, we train a model that jointly learns a 2D sprite dictionary, capturing recurring visual elements in an image collection, and explains each input frame as a combination of these potentially transparent sprites. Whereas standard CNN-based generators hide their feature representation in their intermediate layers, our model wears its representation "on its sleeve": by explicitly compositing sprites from its learnt dictionary onto a background canvas, rather than synthesizing pixels from hidden neural features, it provides a readily-interpretable visual representation.

Our contributions include the following:

- We describe a grid-based anchor system along with a learned dictionary of textured patches (with transparency) to extract a sprite-based image representation.
- We propose a method to learn the patch dictionary and the grid-based representation jointly, in a differentiable, end-to-end fashion.
- We compare to past work on learned disentangled graphics representations for video games.
- We show how our method offers promising avenues for further work towards identifying visual patterns in more complex data such as natural images and video.

35th Conference on Neural Information Processing Systems (NeurIPS 2021).

# 1 Related Work

Decomposing visual content into semantically meaningful parts for analysis, synthesis, and editing is a long-standing problem. We review the most closely related work.

**Layered decompositions.** Wang and Adelson [54] decompose videos into layers undergoing temporally-varying warps for compression. Similarly, Flexible Sprites [24] and Kannan et al. [27] represent videos with full-canvas semi-transparent layers to facilitate editing. Like Flexible Sprites, we adopt translation-only motion but restrict transformations to small neighborhoods around anchors, making inference tractable with many ($\geq 100$) sprites. Other methods decompose videos with moving subjects, such as humans, into independent layers, enabling matting [41] and retiming of individual actions [40]; unlike sprite-based techniques, motion and appearance are not disentangled. Sbai et al. [48] use a layered representation as inductive bias in a GAN with solid colored layers. Automatic decompositions into "soft layers" according to texture, color, or semantic features have been used in image editing [1, 2]. Gandelsman et al. [12] use deep image priors [52] to separate images into layer pairs. Huang and Murphy [19] introduce a recurrent architecture to output multiple layers sequentially. Reddy et al. [46] discover patterns in images via differentiable compositing.

**Interpretable generators for neural synthesis.** Neural networks improve the fidelity and realism of generative models [14, 28] but limit control and interpretability [5, 6, 8, 16]. Several works explore interpretability using differentiable domain-specific functions. Hu et al. [18], Li et al. [31] constrain the generator to sets of parametric image operators. Mildenhall et al. [44] use a ray-marching prior and rendering model to encode a radiance field for novel view synthesis. Neural textures [51] replace RGB textures on 3D meshes with high-dimensional features. Rendering under new views enables view-consistent editing. Lin et al. [33] use spatial transformers in their generator to obtain geometric transformations. We synthesize frames by compositing 2D sprites undergoing rigid motions, enabling direct interpretation and control over appearance and motion.

**Object-centric representations.** Our learned sprites reveal, segment, and track object instances. Similarly, Slot Attention [37] extracts object-centric compositional video representations. However, our sprites are interpretable—motion and appearance are direct outputs—and our model scales to more objects per scene. SCALOR [23] handles up to 100 instances but does not produce a common dictionary or handle diverse sprites. While SPACE [34] decomposes images into object layers, it tends to embed sprites in the background, providing no control. Our method achieves a higher IoU of recurring sprite patterns (see §3.1). Stampnet [53] discovers and localizes objects but focuses on simpler, synthetic datasets. MONet [7] decomposes images into multiple object regions using attention. Earlier attention mechanisms leverage pattern recurrence [9, 30] and motion cues [11] to identify individual objects. Recent works use parametric primitives as image building blocks [32, 49].

Applying our sprite decompositions to video games, we can learn about dynamics and gameplay, benefiting downstream agents [17, 26] and aiding content-authoring for research and game development, as in Procedural Content Generation [50]. GameGAN [29] synthesizes new frames from controller input. They split rendering into static and dynamic components but render full frames, without factorization into parts. Their generator is difficult to interpret: appearance and dynamics are entangled within its parameters.

**Compression.** Appearance consistency and motion compensation are central to video compression [4, 38, 42]. We model videos as compositions of moving sprites, factoring redundancy in the input. This draws inspiration from works like DjVu [15] and Digipaper [20], which compress scanned documents by separating them into a background layer and foreground text. Image epitomes [25] summarize and compress image shape and appearance into a miniature texture. Our sprite dictionary fills a similar role, providing superior editing control.

# 2 Method

We start with an input sequence of $n$ RGB frames $\{I_1, \ldots, I_n\}$ with resolution $w \times h$. Our goal is to decompose each frame $I_i \in \mathbb{R}^{3 \times w \times h}$ into a set of possibly overlapping sprites, organized into $\ell$ depth layers, selected from a finite-size dictionary. The dictionary is a collection of trainable latent codes $\{z_1, \ldots, z_m\}$ that are decoded into RGBA sprites using a neural network generator (§2.1).

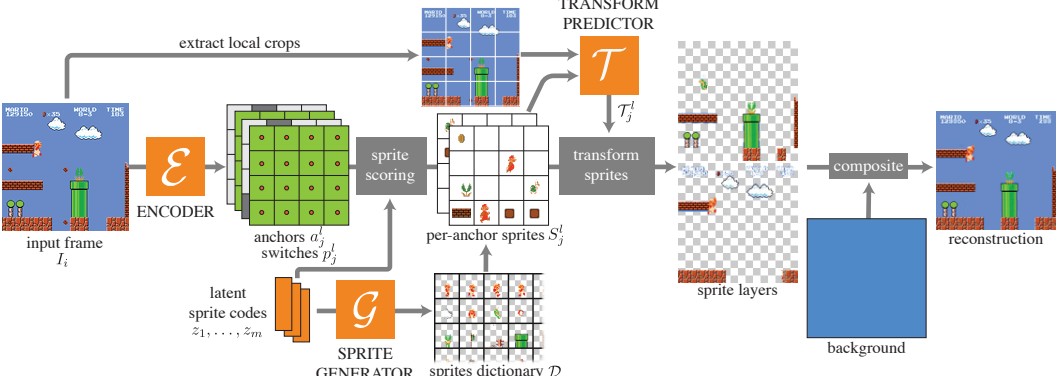

Figure 1: Overview. We jointly learn a sprite dictionary, represented as a set of latent codes decoded by a generator, as well as an encoder network that embeds a frame into a grid of latent codes, or *anchors*. Comparing anchor embeddings to dictionary codes lets us assign a sprite to each grid cell. Our encoder also outputs a binary switch per anchor to turn sprites on and off. After compositing, we obtain a reconstruction of the input. Our self-supervised training optimizes a reconstruction loss.

Our training pipeline is illustrated in Figure 1. We first process each input frame with a convolutional encoder to produce $\ell$ grids of feature vectors, one grid per depth layer (§2.2). The grids are lower resolution than the input frame, with a downsampling factor proportional to the sprite size. We call the center of each grid cell an *anchor*. We compare each anchor's feature vector against the dictionary's latent codes, using a softmax scoring function, to select the best matching sprite per anchor (§2.3). Using our sprite generator, we decode each anchor's matching sprite. This gives us a grid of sprites for each of the $\ell$ layers. To factorize image patterns that may not align with our anchor grid, we allow sprites to move in a small neighborhood around anchors (§2.4). We composite the layers from back to front onto the output canvas to obtain our final reconstruction (§2.5). Optionally, the background is modeled as a special learnable sprite that covers the entire canvas.

We train the dictionary latent codes, frame encoder, and sprite generator jointly on all frames, comparing our reconstruction to the input (§2.6). This self-supervised procedure yields a representation that is sparse, compact, interpretable, and well-suited for downstream editing and learning applications.

## 2.1 Dictionary and sprite generator

The central component of our representation is a global *dictionary* of $m$ textured patches or sprites $\mathcal{D} = \{P_1, \ldots, P_m\}$, where each $P_i \in \mathbb{R}^{4 \times k \times k}$ is an RGBA patch. Our sprites have an alpha channel, which allows them to be partially transparent, with possibly irregular (i.e., non-square) boundaries. This is useful for representing animations with multiple depth layers and also allows to learn sprites smaller than their maximal resolution, if necessary, by setting alpha to zero around the boundary. The dictionary is shared among all frames; we reconstruct frames using only sprites from the dictionary.

Instead of optimizing for RGBA pixel values directly, we represent the dictionary as a set of trainable latent codes $\{z_1, \ldots, z_m\}$, with $z_i \in \mathbb{R}^d$. We decode these codes into RGBA sprites using a fully-connected sprite generator $P_i = \mathcal{G}(z_i)$. This latent representation allows us to define a similarity metric over the latent space, which we use to pair anchors with dictionary sprites to best reconstruct the input frame (§2.3). At test time, we can forego the sprite generator and edit the RGBA sprites directly. Unless otherwise specified, we set latent dimension to $d = 128$ and patch size to $k = 32$.

We randomly initialize the latent codes from the standard normal distribution. Our sprite generator first applies zero-mean unit-variance normalization—Layer Normalization [3], without an affine transformation—to each latent code $z_i$ individually, followed by one fully-connected hidden layer with $8d$ features, Group Normalization [55], and ReLU activation. We obtain the final sprite using a fully-connected layer with sigmoid activation to keep RGBA values in $[0, 1]$. Latent code normalization is crucial to stabilize training and keep the latent space in a compact subspace as the optimization progresses. See §3.3 for an ablation study of this and other components.

## 2.2 Layered frame decomposition using sprite anchors

We seek a decomposition that best explains each input frame using dictionary sprites. We exploit translation invariance and locality in our representation; our sprites are "attached" to a regular grid of reference points, or *anchors*, inspired by [13, 47]. Each anchor has at most one sprite; we call it *inactive* if it has none.

We give the sprites freedom of motion around their anchors to factorize structures that may not be aligned with the anchor grid. This local—or, Eulerian—viewpoint makes inference tractable and avoids the pitfalls of tracking the global motion of *all* the sprites across the canvas (a Lagrangian viewpoint). To enable multiple layers with sprite occlusions, we output $\ell > 1$ anchor grids for each frame ($\ell = 2$ in our experiments). Figure 2 illustrates our layered anchor grids and local sprite transformations.

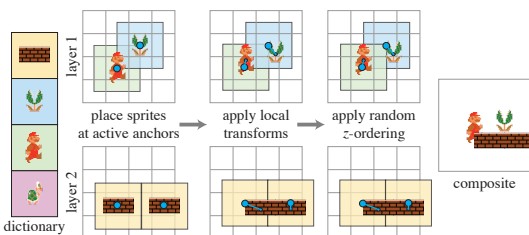

Figure 2: Layered sprite decomposition with local anchors. We assign at most one sprite per anchor and predict a local transformation of each placed sprite around its anchor. To allow for occlusions between sprites, we use multiple sprite layers, which we compose back to front to obtain the final image.

We use a convolutional encoder $\mathcal{E}$ to map the $w \times h$ RGB frame $I_i$ to $\ell$ grids of anchors, with resolution $\frac{2w}{k} \times \frac{2h}{k}$. Each anchor $j$ in layer $l$ is represented by a feature vector $a_j^l \in \mathbb{R}^d$ characterizing local image appearance around the anchor and an active/inactive switch probability $p_j^l \in [0,1]$. Our frame encoder contains $\log_2(k) - 1$ downsampling blocks, which use partial convolutions [36] with kernel size 3 and stride 2 (for downsampling), Group Normalization, and Leaky ReLU. It produces a tensor of intermediate features for each layer, which are

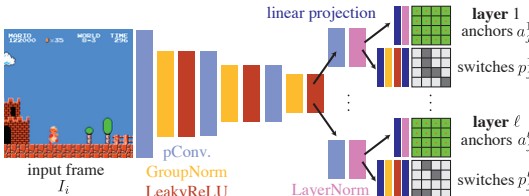

Figure 3: Encoder architecture.

normalized with LayerNorm. From these, we obtain the anchor switches with an MLP with one hidden layer of size $d$ followed by Group Normalization and Leaky ReLU. We get anchor features using a linear projection followed by LayerNorm. The encoder architecture is illustrated in Figure 3.

## 2.3 Per-anchor sprite selection

Once we have the layered anchor grids for the input frame, we need to assign sprites to the active anchors. We do this by scoring every dictionary element $i$ against each anchor $j$ at layer $l$, using a softmax over dot products between dictionary codes and anchor features:

$$s_{ij}^l = \frac{\exp\left(a_j^l \cdot z_i / \sqrt{d}\right)}{\sum_{k=1}^m \exp\left(a_j^l \cdot z_k / \sqrt{d}\right)}. \tag{1}$$

Recall that both the anchor features and dictionary latent codes are individually normalized using a Layer Normalization operator. Restricting both latent spaces to a compact subspace helps stabilize the optimization and avoid getting stuck in local optima. During training, each anchor's sprite is a weighted combination of the dictionary elements, masked by the anchor's active probability:

$$S_j^l = p_j^l \sum_{i=1}^m s_{ij}^l P_i. \tag{2}$$

This soft patch selection allows gradients to propagate to both dictionary and anchor features during training. Except for natural image and video datasets, at test time, we use hard selections, i.e., for each anchor, we pick the sprite ($S_j^l := P_i$) with highest score $s_{ij}^l$ and binarize the switches $p_j^l \in \{0, 1\}$.

## 2.4 Local sprite transformations

In real animations, sprites rarely perfectly align with our regular anchor grid, so, to avoid learning several copies of the same sprites (e.g., all sub-grid translations of a given image pattern), we allow sprites to move around their anchors. In our implementation, we only allow 2D translations of up to $1/2$ the sprite size on each side of the anchor, i.e., $\mathcal{T}_j^l = (x_j^l, y_j^l) \in [-k/2, k/2]^2$.

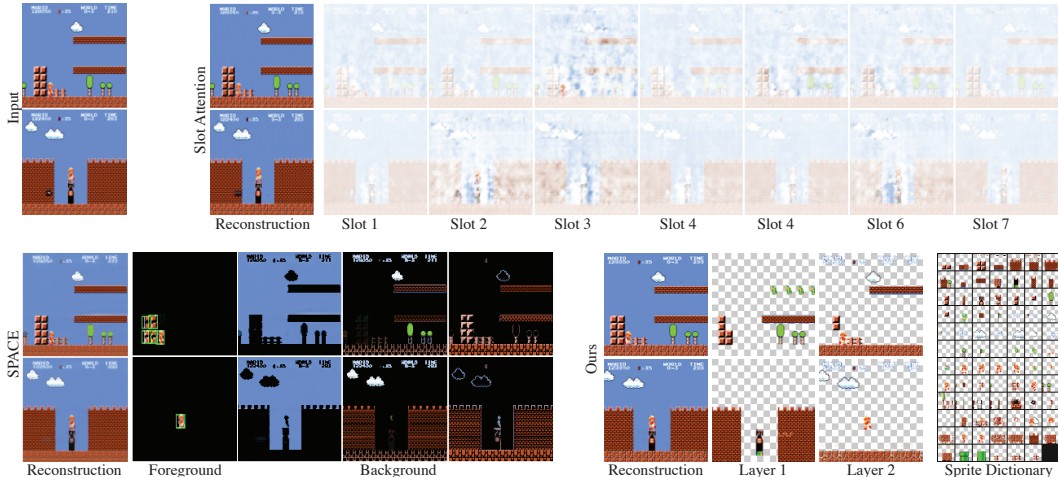

Figure 4: Comparison to SPACE [34] and Slot Attention [37]. While all three methods obtain good reconstructions, SPACE only recognizes a few sprites, and Slot Attention does not yield a meaningful decomposition. We decompose the entire foreground and learn a dictionary.

We use a convolutional network to predict the translation offsets from the anchor's sprite and a crop of the input frame centered around the anchor, with identical spatial dimensions. This network follows the architecture of $\mathcal{E}$ followed by an MLP with a single hidden layer of size $d$, Group Normalization, and Leaky ReLU. Specifically, we concatenate the image crop and the anchor's sprite $S_j^l$ along the channel dimension and pass this tensor through this network to obtain the $x_j^l$ and $y_j^l$ offsets. An output layer projects to two dimensions (horziontal and vertical shift) and applies $\tanh$ to restrict the range. We apply the shifts to the patches using a spatial transformer [21].

## 2.5 Compositing and reconstruction

Each anchor in our layered representation is now equipped with a sprite $S_j^l$ and a transformation $\mathcal{T}_j^l$. For each layer $l$, we transform the sprites in their anchor's local coordinate system and render them onto the layer's canvas, initialized as fully transparent. Because of the local transformation, neighboring sprites within a layer may overlap. When this happens, we randomly choose an ordering, as in Figure 2. This random permutation encourages our model to either avoid overlapping sprites within the same layer or make the sprite colors agree in the overlap region, since these are the only two options that yield the same rendering regardless of the random $z$-ordering. Note that sprites on *distinct layers* are not shuffled. The shuffling prevents the network from abusing the compositing to cover patches with others from the same layer.

We optionally learn a background texture to capture elements that cannot be explained using sprites. This can be thought of as a special patch of resolution greater than that of a single frame. For each frame, we learn a (discrete) position offset in the background from which to crop. We represent these offsets as discrete pixel shifts using a softmax classification (independently for each spatial dimension). We found this encoding better behaved than using a continuous offset with a spatial transformer—the discrete encoding allows the gradient signal to propagate to all shifts rather than the weak local gradient from bilinear interpolation (see §3.3 for an ablation). We combine the background and sprite layers via standard alpha compositing [45]. Figure 8 shows a learned background.

In some experiments, we use a simpler background model: a fixed solid color, determined by analyzing the data before training. In this variant, we sample 100 random frames, cluster the pixel values into 5 clusters using $k$-means, and choose the largest cluster center as the background color.

## 2.6 Training procedure

Our pipeline is fully differentiable. We train the latent codes dictionary, sprite generator, frame encoder, transformation predictor, and background layer jointly, minimizing $L_2$ distance between our reconstructions and ground truth frames. We also employ two regularizers: a Beta distribution prior

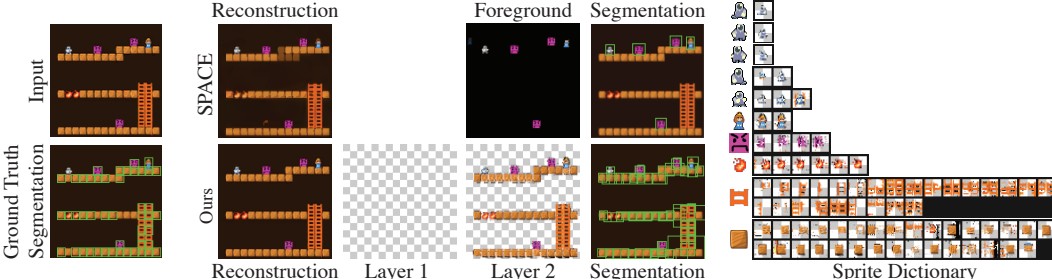

Figure 5: Qualitative comparison to SPACE [34] on the synthetic game dataset. We show ground truth sprite segmentations as well as those obtained from the learned SPACE foreground and from our learned sprites. While SPACE only learns several of the sprites, we reconstruct the entire foreground using our dictionary.

on switches and dictionary element scores favors values close to 0 or 1, and an $L_1$ loss on switches favors a sparser solution without superfluous patches. Our final loss function for a single input is:

$$\mathcal{L}(\cdot) = \tfrac{1}{wh}\|O{-}I\|_2^2 + \tfrac{k^2}{4\ell wh}\sum_{l=1}^{\ell}\sum_{j=1}^{\frac{2w}{k}\times\frac{2h}{k}}\left[\lambda_{\text{Beta}}\left(\tfrac{1}{m}\sum_{i=1}^{m}\text{Beta}(2,2)(s_{ij}^l){+}\text{Beta}(2,2)(p_j^l)\right){+}\lambda_{\text{sparse}}|p_j^l|\right], \quad (3)$$

where $O$ is the result of compositing the background and sprite layers; we optimize $\{s_{ij}^l\}$, $\{p_j^l\}$, and $O$. We set $\lambda_{\text{sparse}} = 0.005$ and train for 200,000 steps ($\sim$20 hours) with $\lambda_{\text{Beta}} = 0.002$ and finetune for 10,000 steps with $\lambda_{\text{Beta}} = 0.1$. For natural images and video, we set $\lambda_{\text{Beta}} = 0$. We use the AdamW [39] optimizer on a GeForce GTX 1080 GPU, with batch size 4 and learning rate 0.0001, except for the background module (learning rate 0.001 when used).

## 3  Experimental Results

We evaluate our self-supervised decomposition on several real (non-synthetic) datasets, compare to related work, and conduct an ablation study. In figures, we use a checkerboard to show transparency. Dictionary order is determined by sorting along a 1-dimensional $t$-SNE embedding of the sprite latent codes. We find this ordering tends to group semantically similar sprites, making the dictionary easier to interpret and manipulate. While our models are trained with a dictionary of 150 patches, not all patches end up being used; we only show the used patches.

### 3.1  Comparisons

While to our knowledge no prior works target differentiable unsupervised sprite-based reconstruction, we compare to two state-of-the-art methods that obtain similarly disentangled representations.

In Figure 4, we compare to SPACE [34] and Slot Attention [37]. The former decomposes a scene into a foreground layer consisting of several objects as well as a background, segmented into three layers. The latter deconstructs a scene into discrete "slots." We train both methods to convergence using their default parameters. While both reconstruct the input frames faithfully, SPACE only recognizes a few sprites in its foreground layer, and Slot Attention does not provide a semantically meaningful decomposition. In contrast, not only does our method model the entire scene using learned sprites, but also it factors out the sprites to form a consistent, sparse dictionary shared for the entire sequence.

Additionally, we evaluate on a synthetically-generated sprite-based game from [10], which is made of sprites on a solid background. We compare quantitatively to SPACE in Figure 6 and show qualitative results in Figure 5. Since we have a ground truth segmentation of each scene into sprites, we compute a matching between learned dictionary patches and sprites by associating each dictionary patch with the sprite that it most frequently overlaps across the dataset. We visualize dictionary patches next to their respective sprites. We also use this labeling to compute segmentation metrics. In particular, we report mean IoU in the multiclass case (where each sprite is a distinct class) as well is in the binary case (foreground/background). Because SPACE does not learn a common dictionary, we are unable to

obtain a labeling for its foreground elements and, consequently, cannot evaluate its multiclass metric. For the binary metric, we obtain a significantly higher value, since SPACE defers many sprites to the background, whereas our method learns the sprites as dictionary elements.

To show that our model learns more than simple motion features, we also compare to two conventional (non-learning) baselines. In Figure 7(a), we compare a segmentation of a frame obtained by clustering optical flow directions using $k$-means (inspired by Liu et al. [35]) to one generated using our learned decomposition. The flow-based approach is unable to capture many of the details in the frame. In (b), we show the normalized dictionary obtained using an online dictionary learning method [43]. Because this method does not have the inductive biases of our model, the resulting dictionary is not easily interpretable or editable.

Input  Flow-Based  Ours

(a) Segmentation

(b) Online Dictionary Learning

Figure 7: Comparison to conventional baselines.

## 3.2 Sprite-based game deconstruction

We train on Fighting Hero (one level, 5,330 frames), Nintendo Super Mario Bros. (one level, 2,220 frames), and ATARI Space Invaders (5,000 frames). We use patch size $k = 32$ for Mario and Fighting Hero and $k = 16$ for Space Invaders. For Fighting Hero, we learn a background, as described in §2.5.

The sprites, background, and example frame reconstructions are shown in Figure 8. Our model successfully disentangles foreground from background and recovers a reasonable sprite sheet for each game. Having reverse-engineered the games, we can use the decomposition for applications like editing. In Figure 9, we demonstrate a GUI that allows the user to move sprites around the screen.

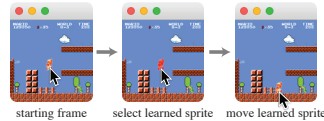

starting frame    select learned sprite    move learned sprite

Figure 9: Editing GUI.

## 3.3 Ablation Study

We show an ablation study on the Mario data. We train our full model, one with smaller $16 \times 16$ patches, another with larger $64 \times 64$ patches, a model with a smaller dictionary (25 elements), a model without LayerNorm, and one where we use a straight-through estimator [22] to learn discrete switches $p_j^l$ in lieu of Beta regularization. We train each model with five random seeds and report the reconstruction PSNR means and standard deviations in Figure 10. This experiment verifies the importance of LayerNorm in our architecture and shows that the straight-through trick is ineffective in our setting. Though the smaller patches model achieves slightly higher mean PSNR than our full model, more of the sprites are split across dictionary patches (Figure 11), illustrating how the patch size choice sets an inductive bias for our decomposition.

| Model | PSNR |
|---|---|
| Smaller patches | $28.85 \pm 0.95$ |
| Full | $28.04 \pm 0.72$ |
| No LayerNorm | $26.05 \pm 0.45$ |
| Smaller dictionary | $23.80 \pm 1.38$ |
| Larger patches | $23.63 \pm 1.05$ |
| Straight-through switches | $22.15 \pm 0.25$ |

Figure 10: Ablation study on Mario data across five random seeds.

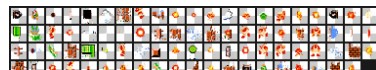

Figure 11: Mario dictionary with $16 \times 16$ patches.

We also justify our choice for learning background shifts via classification (§2.5) rather than regression, i.e., using spatial transformers. Figure 12 shows the background learned using a spatial transformer. In contrast to our full model (Figure 8), the original background is not discovered, and most of the canvas is unused. We suspect that this is due to lack of gradient signal from background pixels that do not get rendered at each training step.

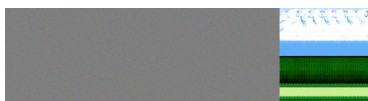

Figure 12: Background learned with spatial transformer.

| Method | Reconstruction PSNR | Mean IoU (multiclass) | Mean IoU (binary) |
|---|---|---|---|
| SPACE | 31.9 | - | 0.0361 |
| Ours | **38.54** | **0.6497** | **0.7352** |

Figure 6: Quantitative comparison to SPACE [34]. We report PSNR to evaluate overall reconstruction quality as well as mean IoU for multiclass and binary foreground/background segmentation problems. Our method recognizes significantly more sprites than SPACE, resulting in higher mean IoU.

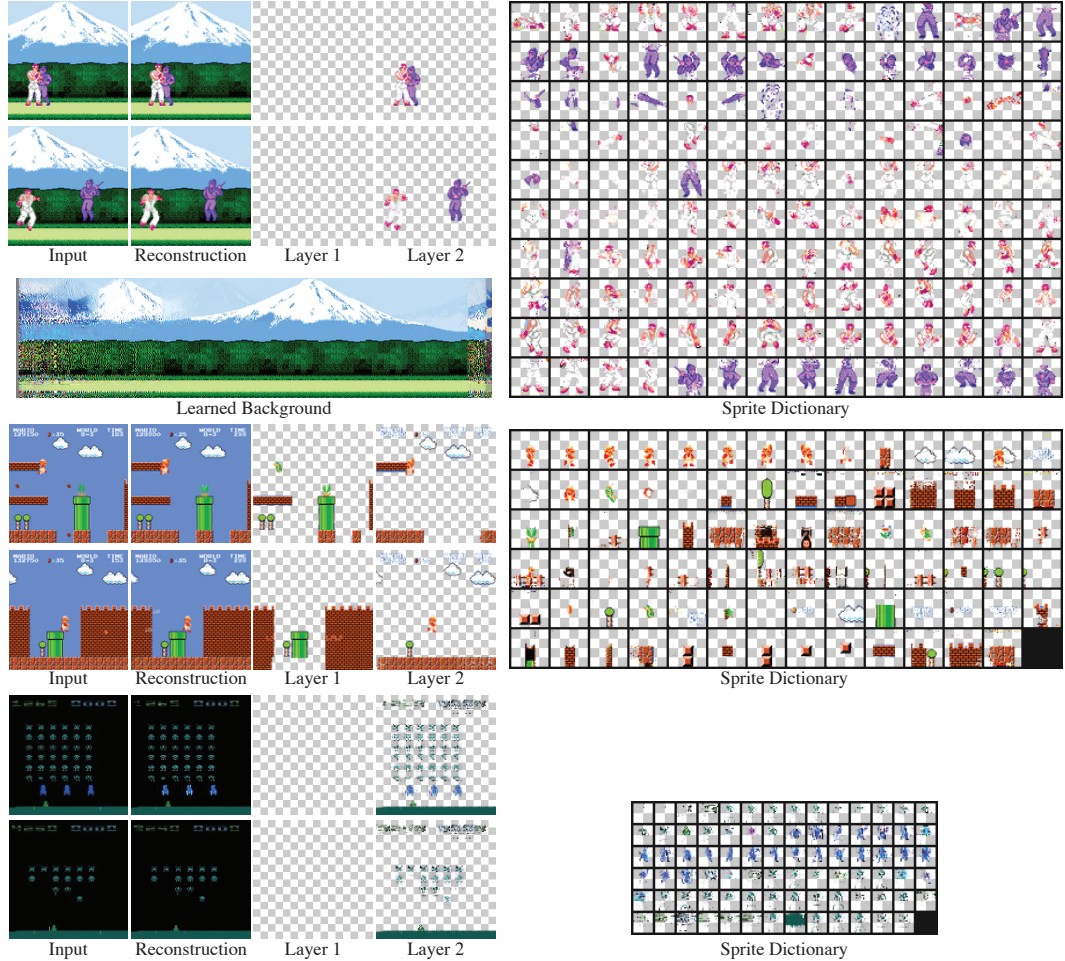

Figure 8: Sprite-based game decompositions. Our self-supervised technique recovers a compact dictionary of semantically meaningful sprites representing characters (or their body parts) and props. The first example shows our learned background texture; the others use a solid color as background.

### 3.4 Future Directions and Limitations

While our method is designed with sprite-based animation in mind, it can generalize to natural images and videos. An exciting direction for future work is to incorporate more expressive transformations so as to discover recurring content in generic videos. Here, we obtain preliminary results using our approach and achieve interesting decompositions even without modifications to our sprite-based model.

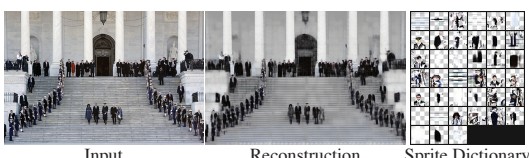

Figure 13: We factorize the recurring elements of a single photograph into a compact dictionary.

In Figure 14, we show results on a tennis video (4,000 frames). The model learns parts of the player's body (head, limbs, shirt, etc.) as sprites and captures most of the tennis court in the learned background. By simply selecting the player sprites in the dictionary, we segment the entire video clip.

Our model can also discover recurring patterns in a single natural image. We train on random crops of a $768 \times 512$ photograph from the 2013 US Presidential Inauguration,[1] which contains many repeating elements such as stairs, columns, and people. With a dictionary of 39 $32 \times 32$ sprites (39,936 pixels), we recover much of the detail of the original 393,216 pixels.

---

[1] AP Photo/Cliff Owen

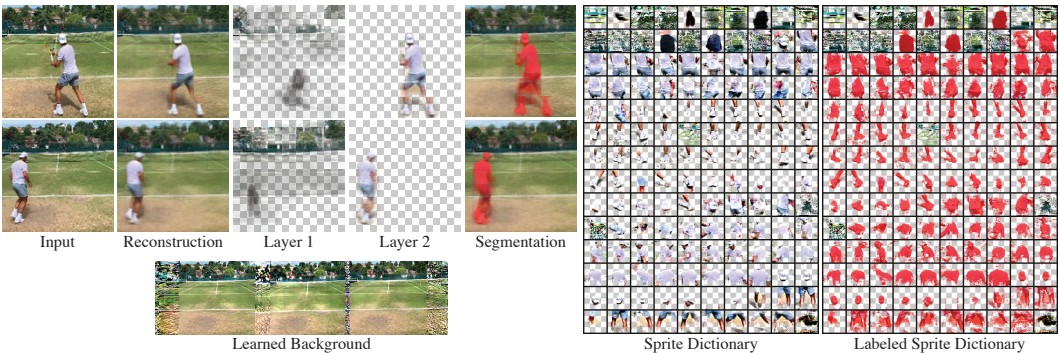

Figure 14: Segmentation of natural videos. Despite its simplistic motion and appearance model, our approach can be applied to real-world videos. By selecting the few sprites corresponding to the tennis player, we can quickly obtain a segmentation of the full video sequence.

Figure 15: Reconstruction of a scanned text excerpt, illustrating some limitations of our method. Because letters are densely packed within the text and lack motion cues, our model learns sprites comprising more than a single glyph and suffers from high reconstruction error.

We demonstrate further limitations of our approach by applying it to automatic font discovery. We train on random $128{\times}128$ crops of six scanned pages of *Moby Dick*, each of approximately $500{\times}800$ resolution. Figure 15 shows an input text excerpt, our reconstruction, and the learned dictionary.

This dataset differs significantly from our other testing datasets. Each input frame consists of many densely packed sprites ($\sim 100$ glyphs in each $128{\times}128$ crop), and many individual glyphs consist of smaller repeating elements. We hypothesize that because of these issues, combined with a lack of motion cues between frames, we do not achieve a perfect reconstruction, learning certain sprites with multiple glyphs and others with just partial glyphs. Incorporating priors tailored to regularly structured and dense data like text is a direction for future research.

## 4 Conclusion

We present a self-supervised method to jointly learn a patch dictionary and a frame encoder from a video, where the encoder explains frames as compositions of dictionary elements, anchored on a regular grid. By generating layers of alpha-masked sprites and predicting per-sprite local transformation, we recover fine-scale motion and achieve high-quality reconstructions with semantically meaningful, well-separated sprites. Applied to content with significant recurrence, our approach recovers structurally significant patterns.

Understanding recurring patterns and their relationships is central to machine learning. Learning to act intelligently in video games or in the physical world requires breaking experiences down into elements between which knowledge can be transferred effectively. Our sprite-based decomposition provides an intuitive basis for this purpose. In this work, we focus on a simplified video domain. In the future, we would like to expand the range of deformations applied to the learned dictionary elements, such as appearance or shape changes. Our work opens significant avenues for future research to explore recurrences and object relationships in more complex domains.

## Acknowledgements

The MIT Geometric Data Processing group acknowledges the generous support of Army Research Office grants W911NF2010168 and W911NF2110293, of Air Force Office of Scientific Research award FA9550-19-1-031, of National Science Foundation grants IIS-1838071 and CHS-1955697, from the CSAIL Systems that Learn program, from the MIT–IBM Watson AI Laboratory, from the Toyota–CSAIL Joint Research Center, from a gift from Adobe Systems, from an MIT.nano Immersion Lab/NCSOFT Gaming Program seed grant, and from the Skoltech–MIT Next Generation Program. This work was also supported by the National Science Foundation Graduate Research Fellowship under Grant No. 1122374.

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
