# OpenReview forum: "MarioNette: Self-Supervised Sprite Learning"
_NeurIPS.cc/2021/Conference — NeurIPS 2021 Poster_

### Official Review · Reviewer_dp3z · 2021-07-16

**Rating:** 6
**Confidence:** 4

**Summary:**

The paper presents a framework for unsupervised disentanglement of images from 2d games into sprites. The framework consists of learning a sprite dictionary, predicting corresponding sprites based on the information from the frame and rendering these sprites. The framework is supervised with reconstruction loss plus some regularization.

Overall the solution makes sense and seems well-thought, however the task 2d game decomposition is not that broad.


**Ethical Concerns:**

There is no ethical concerns.

**Limitations And Societal Impact:**

Limitations is discussed in the paper and in the main review. There is no negative impact.

**Main Review:**

Positive:
- The solution to the problem makes sense and seems well thought.
- The paper contains an extensive evaluation and ablation study.
- The paper tries to apply the proposed method for real world data, e.g. scanned text, president photos and tennis matches.

Negative:
- The paper considers a 2d game decomposition task in computer vision fashion, however it is questionable if this task has any practical usefulness. Indeed one could use game sprites from game source code and similarly took their arrangement from there.

- The proposed method is very game specific, each game it requires to tune many parameters: depending on the size of sprites in the game the size of the grid need to be adapted; background model need to be adapted as well, for example in the game like Mario separate background for each lever need to learned (here only one lever is considered); number of layers need to selected if there are more than 2 layers in the game.

Questions:
- In Fig 8. (Mario) The "layer 2" contains some scores and letters renderings which are practically unreadable, however when they are used in reconstruction they look almost perfect, how did this happen? Is the layer just overplayed with the background for rendering or some other post processing is applied?

- For "Transform Predictor" why only shifts are considered, is it possible to use arbitrary affine transformations?

Some minor issues:
- Figure 10 and Figure 6 should be Tables 10 and Table 6.




**Time Spent Reviewing:**

4h

---

> ### Author Response · Authors · 2021-08-10
> **Response**
>
> We thank the reviewer for their feedback and are glad that they appreciate our solution and its evaluation. Below, we clarify some questions raised in the review, which can be easily addressed in the camera-ready version. If there are any concerns left unaddressed by our rebuttal, please let us know, and we will gladly follow up during the discussion period.
>
> ### Usefulness
> While the main datasets we consider have publicly-available ground truth decompositions, our paper motivates a broad class of applications and methods. As we show in our preliminary results in Section 3.4, our framework offers promising applications to natural images and scanned text, for which ground truth is not readily available. Learning explicit decompositions of visual content is crucial for improving machine learning models in several areas (interpretability, control, editability, etc.). This is a budding research topic, as evidenced by the quality of current state-of-the-art approaches, for which sprite-based animation and game content remain a challenging testbed.
>
> ### Choice of hyperparameters
> While our model does allow several hyperparameters to be determined prior to training, these are not critical for the success of the model and are intuitive to select, not requiring precise tuning or brute-force parameter sweep.
> - Choosing a sprite size allows the user to inject implicit bias for the granularity of sprites considered. As we show in Figure 11, choosing the “wrong” sprite size still yields a good reconstruction and decomposition.
> - While the number of layers can also be selected, using two layers is a good default choice. The number of layers just needs to be no smaller than the maximal number of overlapping sprites. In all our experiments, we did not need additional layers.
>
> ### Post-processing
> The output images are produced using standard back-to-front alpha compositing; no post-processing is performed. Indeed, the text is less legible in the sprite sheet than in the final reconstruction because it is split across several sprites, and some of the background color leaks into the learned patches. It is also difficult to discern the white text on the white-gray grid background in our visualization, which we can improve in the final revision. Unlike the other sprites we recover, the text is consistently located in the same position and on the same background across all frames in the dataset, giving the model no motion cues for learning a higher-quality decomposition.
>
> ### Arbitrary spatial transformations
> We restrict the class of allowed transformations to shifts because this model best aligns with our main experimental setting (sprite-based games). An exciting direction for future work is expanding the class of learnable transformations, e.g., to arbitrary affine transformations, or even further, e.g., allowing appearance variations or non-rigid transformations of the “sprites.”

---

> > ### Comment · Reviewer_dp3z · 2021-08-29
> > **Responce to authors**
> >
> > The authors address most of my minor concerns that I had, so I prefer to keep my original score since. Except the "Arbitrary spatial transformations", here I disagree because many sprite-based games contain rotated versions of the original sprites. For example in Contra, when main character is jumping, or bullets in different other shooting games.

---

### Official Review · Reviewer_JHYp · 2021-07-16

**Rating:** 7
**Confidence:** 5

**Summary:**

The authors propose a generative layered model of images based on sprites — the idea is that there is a dictionary of (possibly transparent) sprites which are arranged on layers which are, together with an optionally learned background layer, composited back-to-front to form a given image.  The task is to learn the sprite dictionary and their poses and depth ordering on a given frame.  The paper shows successful results on frames from Atari/SNES era video games (which actually did use sprites) and some intriguing (but very much preliminary) one-off results on real images, training on patches sampled at random from those images.


**Limitations And Societal Impact:**

Yes

**Main Review:**

I like several things about this paper. First, there is pretty good execution in that technically the model make makes sense and these models are never easy to get right (it’s often easy for these models to “cheat”, e.g. by putting all of their information on one layer). It also seems like a clearly novel model and also outperforms some natural competitors at least on old video game data.

On the other hand the main weakness is that the proposed model certainly does not seem terribly general — though it is interesting that the authors are able to show that it can be a generic dictionary learning approach (even if not a very good one).

Overall I’m a bit biased in favor of this paper because the model is elegant and thought provoking so I would like to see it published, with the admission that the motivation for working on such a model is not particularly compelling and the application space seems narrow.

Some more detailed comments/questions:
* Why partial convolutions? I was not familiar with this concept and looked at the reference, but it was not clear what was used as the mask for the partial convolutions in this paper and why it was necessary (is there an ablation for this?)
* In general, how does one decide on background canvas size?
* I notice that motion cues are referenced a few times in paper, but don’t seem to actually be used by model? It *does* makes a lot of sense to assume that sprites move smoothly across the background --- and perhaps have a learnable transition model through the sprites to reflect appearance change of an object.
* Other related work ( recent layered decomposition models based on similar alpha compositing and reconstruction assumptions) that I recommend adding:
 ** Omnimatte (https://omnimatte.github.io/)
 ** Multiplane images (e.g., https://augmentedperception.github.io/deepview/)
 ** Efficient inference in occlusion-aware generative models of images (https://arxiv.org/abs/1511.06362)



**Time Spent Reviewing:**

2

---

> ### Author Response · Authors · 2021-08-10
> **Response**
>
> We thank the reviewer for their feedback and are glad that they appreciate the elegance and novelty of our model, noting that it outperforms natural competitors. Below, we clarify some questions raised in the review, which can be easily addressed in the camera-ready version. If there are any concerns left unaddressed by our rebuttal, please let us know, and we will gladly follow up during the discussion period.
>
> ### Generalizability
> Expanding our framework to work robustly on natural videos is an exciting direction for future research. The figures in Section 3.4 demonstrate preliminary results; a richer image formation model (e.g., including illumination variations) would likely be needed to achieve photo-realistic reconstructions of complex videos.
>
> ### Use of partial convolutions
> We use partial convolutions as a padding scheme (see “Partial Convolution based Padding” by Liu et al.; we will add this citation). This gives an elegant way to do convolutions without prescribing boundary conditions. In practice, this approach yielded better experimental results than conventional padding options. We suspect this is because using other padding types (e.g., zero-padding) adds an undesired bias to the latent codes learned at the boundary anchors (compared to those for the interior anchors). This is because the receptive field of the boundary patches always includes the padded area, whereas that of the interior anchors does not. Using partial convolutions avoids this issue.
>
> ### Choice of background size
> We found that the model is insensitive to the choice of background canvas size---as long as the chosen size is no smaller than the “true” background size, the model learns successfully. In our experience, if an overly small background size is picked, some sprites end up being used to fill in missing background details, but the reconstruction is still successful.
>
> ### Motion cues
> Our model relies on motion cues implicitly during training by virtue of seeing many distinct frames of the animation. The differences betwen frames due to motion provide signal to separate foreground sprites and background. Incorporating this prior more explicitly, e.g., with a temporal continuity regularizer, is an interesting direction for future work.
>
> ### Other
> Thank you for suggesting these references; we will add them to our discussion of related works.

---

> > ### Comment · Reviewer_JHYp · 2021-08-31
> > **Response**
> >
> > Thank you authors - this answers my questions.  Again, I like the paper and will keep my score as is.

---

### Official Review · Reviewer_7ZfW · 2021-07-17

**Rating:** 7
**Confidence:** 5

**Summary:**

This paper proposes a method to disentangle sprite-based video animations into recurring visual elements in a self-supervised manner. Specifically, it decompose a frame into grids and try to reconstruct the frame by drawing elements from a dictionary of discovered sprites for each grid. By training with this reconstruction objective, the model could discover a set of disentangled sprite elements. The authors show the effectiveness of their method both through qualitative discovery results as well as quantitative segmentation metrics. With these results, the authors demonstrate that their method work much better compared to previous methods in terms of discovering these recurring visual elements.


**Ethical Concerns:**

No ethical concerns spotted.


**Limitations And Societal Impact:**

Yes


**Main Review:**

+ The topic of self-supervised discovery of recurring visual elements is very interesting. Though has been explored before, the proposed method tackles this problem with a very explicit and intuitive grid-based modeling approach.
+ The ides is well executed, the qualitative results are convincing to show that the method works pretty well in the video game animation domain. Also, as the quantitative results demonstrated, the proposed method has a large advantage compared to previous methods in visual discovery and disentanglement.
+ The paper is well-written and easy to follow.
+ The authors have promised to release the code and models.
+ I appreciate the discussion on the limitation of the proposed approach.

Regarding the approach
- The biggest concern I have about this work is that, though demonstrated to work very well on video game frames, it's largely unknown how well it can generalize to the domain of real videos. Even with the example shown in Fig. 14, the video is relatively simple and with clean background. It would be nice to see how it works in more challenging scenarios, e.g. evaluate segmentation quality on datasets like DAVIS.
- I understand this paper is trying to make a step towards self-supervised visual discovery and appreciate the choice of video game frames as a controllable and easy-to-source testbed to develop prototypes. With that, I'm not sure what is the utility of the editing GUI demonstrated in Figure 9? Since for video games, we already have the full control of all disentangled elements, why do we need to develop such an application?
- In Eq 3, I'm curious how important are those regularization terms? Does the method learn at all without them?

Typos
- In Figure 3, the bottom-right texts should be "switches p^l_j"?

Overall, I appreciate the direction taken and the approach developed in this paper and believe it would be inspiring for the community.

####### Post rebuttal #######
I thank the authors for their response, they are useful in further clearing some questions I had about the work. I'd like to maintain my score to accept this paper.


**Time Spent Reviewing:**

6

---

> ### Author Response · Authors · 2021-08-10
> **Response**
>
> We thank the reviewer for their feedback and are glad that they appreciate our contributions, noting our paper is easy to follow and well-executed, with convincing qualitative and quantitative results. Below, we clarify some questions raised in the review, which can be easily addressed in the camera-ready version. If there are any concerns left unaddressed by our rebuttal, please let us know, and we will gladly follow up during the discussion period.
>
> ### Generalizability to real videos
> Expanding our framework to work robustly on natural videos is an exciting direction for future research. The figures in Section 3.4 demonstrate preliminary results; our method could be extended with, e.g., more generic transformations to support a broader class of input data. This would, however, require further investigation, beyond the scope of the paper.
>
> ### Editing GUI
> We present the GUI as a proof of concept for a possible content-editing tool using our learning framework. It demonstrates the quality of our automatic sprite decomposition, which enables downstream edits such as sprite replacement, rearrangement, etc. Even though we showcase it on a simple sprite-based game for which ground truth sprites are publicly available, the same GUI could be used to edit more complex data, such as hand-drawn games or pre-digital animations, scanned text, etc.
>
> ### Importance of regularization terms
> While the regularization terms in Equation 3 qualitatively improve the sprite decomposition, training the model without these terms yields equally good reconstruction quality and only slightly worse-looking sprites. In particular, removing the sparsity regularizer introduces some duplicate sprites in the learned dictionary, and removing the Beta regularizer yields minor artifacts when quantizing the learned discrete variables at test time. If requested, we will gladly add an ablation experiment to demonstrate this.

---

### Official Review · Reviewer_vWXs · 2021-07-17

**Rating:** 7
**Confidence:** 3

**Summary:**

The paper introduces a technique for unsupervised / self-supervised decomposition of a scene into an arrangement of layered grids of offset sprites drawn from a simultaneously learned sprite sheet. The main contributions are the network architecture and training objective.

**Ethical Concerns:**

None.

**Limitations And Societal Impact:**

Some limitations of this work are pointed out and visually represented in the Limitations section. The Moby Dick text example is particularly useful because it represents a case where a grid of sprites transformed by only scaling *should* be sufficient for excellent reconstruction (but the demonstrated reconstruction quality is low, presumably because the grid used for analysis does not allow for enough density in the right places, forcing the system to learn character-clusters for which it doesn't have enough capacity).

The significance of transforming sprites with only translation is not well problematized (particularly given the scale/lighting variation seen for objects in the 3d-rooms dataset used to evaluate related work).

**Main Review:**

(An earlier version of this review interpreted the paper as only having effective applications to images that were originally built from a grid of tiles where the ground truth grid size was matched to the grid sized used for analysis or a precise multiple of it. A clarification of the data preprocessing pipeline clarified the situation. Several critical comments that were founded on that observation have been removed here. Main score changed from 3 to 7.)

The abstract suggests “Our framework offers a promising approach for discovering recurring visual patterns in image collections without supervision,” and very good results are shown for Super Mario Bros. It might seem that the technique relies on a careful coherence between the grid structure it uses for analysis and a tile grid being used to generate the original data used for evaluation. However, the fact that a reasonably effective reconstruction is found for tennis while an ineffective reconstruction is found for the Moby Dick text image shows that the situation is not as simple as being specific to tile-oriented images.

The way the this approach "wear its representation 'on its sleeve'" seems to be both a strength and weakness. Reifying the recurring visual patterns as a library of sprite images makes the library easy to audit, but bakes certain features into the pattern representation that we'd normally like representations to abstract over, e.g. robustness to slight variations scale, pose, lighting, etc. Scaling the brightness of an input image to 50% wouldn't effect most other approaches, but it would seem to have a major impact the scene reconstruction ability of this approach. Changing the scale of the images 20% between train and test might also break this approach. These problems do not seem to be unrecoverable within the proposed approach -- the "transform predictor" could be changed to account for more transformations -- but the evaluations performed in the paper don't probe the limits of the translation-only design choice.

Overall, the results *do* show a clear improvement over approaches like SPACE in settings where translation is the only transformation needed. (SPACE was also evaluated on a 3d-rooms dataset where objects varied in scale and lighting.) The state of the art is being advanced in this paper.


**Time Spent Reviewing:**

6

---

> ### Author Response · Authors · 2021-08-10
> **Response**
>
> We thank the reviewer for their feedback. We remain confident that our paper will be a welcome contribution to the NeurIPS community.
>
> Below we address some misunderstandings. If there are any concerns left unaddressed by our rebuttal, please let us know, and we will gladly follow up during the discussion period.
>
> ### Problem statement
> Beyond video games, our paper makes a contribution towards the difficult goal of studying ways to decompose complex high-dimensional visual signals. We choose a simplified image formation model, adapted to sprites common in animation and games, because this setup already proves to be challenging for current state-of-the-art methods (see Section 3.1 for comparisons). We show how this simplified model performs on more complex data like natural images in Section 3.4. These experiments demonstrate the potential of our approach but are only preliminary. A richer image formation model that accounts for changes in illumination, more complex spatial deformations, etc. would be needed but remains an unexplored area for future research.
>
> ### Past work
> It is regrettable that our characterization of prior work came across as reckless. To clarify, sprite sheets are a well known technique in the graphics community that reduces the labor of animation artists compared to manually drawing each frame from scratch. The particular claim that the reviewer mentions requires citation (“By reusing the sprite sheet, authoring new content requires minimal effort…”) describes an idea that has been commonly used in various contexts for decades and does not trace back to a single source; we will gladly add references to computer graphics texts (e.g., "The Art and Science of Computer Animation" [Mealing 1999], "The Commodore 64 in Action" [Novak 1984]) or any other specific references suggested by the reviewer.
>
> ### Motivation in the introduction
> We will make our motivation more clear. Our goal is to produce an **explicit** image decomposition that can be learned from data, which is what we meant by “our model wears its representation ‘on its sleeve’”. This is in contrast to machine learning methods, in particular deep learning approaches, that encode signals into hidden feature representations that are difficult for a human to control or interpret. State-of-the-art work on such explicit representations (which we compare to in Section 3.1), show the difficulty of the task. We will expand on this discussion.
>
> ### Evaluation and challenge of the sprite decomposition task
> We argue that using real unlabeled data from sprite-based games forms a convincing testbed for our method. Note that related work on object-centric representations includes many papers (such as “Object-Centric Learning with Slot Attention” [Locatello 2020], which we compare to) that are validated **exclusively on synthetically-generated [datasets](https://github.com/deepmind/multi_object_datasets)**. While the sprite-based games that we use as our main examples have publicly-available sprite sheets, we show that the closest state-of-the-art approaches still fail at the decomposition task; our work significantly advances the state of the art on this problem.
>
> We also **do not** make use of “perfect knowledge of the sprite size and grid resolution”---our model is not sensitive to the choice of these parameters, as demonstrated in our ablation study (Section 3.3). Additionally, we show preliminary results on data where ground truth values for such parameters simply do not exist (Section 3.4).
>
> ### Applications beyond sprite-based games
> As stated in the paper, our goal is **not** to achieve state-of-the-art results on human pose identification or background matting. Rather, Section 3.4 demonstrates future directions where our framework can be applied. Deconstructing a natural image into sprites can be useful for, e.g., partial recoloring or other content-based edits, and applying our framework to scanned text could be used to generate fonts automatically. Coupled with more sophisticated image formation models, we believe our approach can open up new ways to think about these problems, producing more explicit learned representations.
>
> Existing methods, including VQGANs, do not learn a latent space that can be directly mapped onto the output image using simple spatial transformations. Therefore, their learned representations do not facilitate applications like those mentioned above. We will gladly add comparisons to VQGAN if requested, but it is unclear how their learned representation can be used to generate an editable and explicit sprite dictionary, such as those produced by our method.

---

> > ### Comment · Reviewer_vWXs · 2021-08-18
> > **Best results only demonstrated for multiples of true grid resolution**
> >
> > Regarding my comment on using "perfect knowledge of the sprite size and grid resolution":
> >
> > The Nintendo Entertainment System uses 8x8 pixel sprites at the hardware level. When this paper reports changing varying the patch size k between various power-of-two multiples of this elementary size, we the reader doesn't learn much about the model's ability to work with images that aren't fundamentally composed of tiles tiles of the a-priori known size. Using a value of k that is a multiple of 7 or 9 might show this. Alternatively, scaling the source image up or down by 10% and using the current k values might show this. It is possible that the sub-grid translation system may account for repeated patterns as they go in and out of phase with this differently scaled grid, but is is not obvious what the impact on sprite and reconstruction quality would be. It would be great to see the model generalize (at test time) to slight variations in scale, but it would be already be more convincing to see the model's (training time) reconstruction with grids that aren't perfect integer multiples of 8.
> >
> > Regarding the author's comment "we show that the closest state-of-the-art approaches still fail at the decomposition task [of reconstructing sprite-based game images]; our work significantly advances the state of the art on this problem": The authors are swapping out an interestingly general problem (decomposition of potentially photograph-like images that might be subject to variations in scale/contrast/brightness) with a specific one (recovery of simple sprites from untransformed scenes with an a-propri known grid resolution). The work with which Marionette is compared (such as SPACE) was demonstrated on more difficult problem instances. When Marionette is demonstrated on scenes without the clean grid structure (text and photos), it is not compared with other techniques. It doesn't follow that the state of the art is being advanced here.
> >
> > The authors don't attempt to argue for the value of strong solutions to the more specific problem, so the reader is left to consider the broader problem on which Marionette is not obviously better than past approaches.

---

> > > ### Author Response · Authors · 2021-08-18
> > > **Follow-up**
> > >
> > > This is correct, the NES sprites are indeed of 8x8 and 8x16 resolution. However, the original NES screen is 256x240 pixels (width x height). To produce our dataset, we uniformly rescale each frame so that the shorter side is 128 pixels (i.e., to ~136x128) and extract a 128x128 center crop. Therefore, the true sprites are not power-of-two multiples of our patch size; rather, there is a 240/128=1.875 scale ratio between the underlying “ground truth” sprites and the sprites we learn. This should allay the reviewer’s concern. We will clarify our data preparation description in the paper and can gladly add an ablation study with more rescaling factors of the ground truth (e.g., +/- 10%) to further show that our model does not overfit to an exact size match.
> > >
> > > We would like to point out that 10 of the 11 datasets shown in SPACE are Atari games, which are no more difficult than our main datasets and have less sprite diversity. The 11th dataset, 3D-room, although slightly more complex, can hardly be qualified as realistic. We claim that no current method, including ours, does well on real-world data, but that our method nonetheless significantly improves the state of the art. Specifically, our comparison on sprite-based datasets shows that our method can recover several dozens of unique sprites. In contrast, SPACE and the other baselines we evaluate produce a fairly small number of distinct sprites. We will gladly run SPACE on our real-world data for comparison.
> > >
> > > Finally, unlike any prior work we are aware of, our method produces an explicit, consistent, global patch dictionary that can explain any image in our dataset. This enables applications in which it is possible to interact directly with the disentangled graphical representation for the purpose of analysis, editing, automation, and so on. To the best of our knowledge, such use cases are simply not feasible with the past techniques we compare to.
> > >
> > > Please let us know if there is any additional information we can provide to continue the discussion.

---

> > > > ### Comment · Reviewer_vWXs · 2021-08-19
> > > > **New understanding**
> > > >
> > > > Thank you for clarifying the situation with the use of scaling during preprocessing. This note completely resolves my concern around implicitly feeding the network the ground truth tile size in Mario, and it causes me to reinterpret the work as a whole. I now see how this really is addressing a more general problem without leaning on a cheap trick that works for the particular bias of the evaluation setting.
> > > >
> > > > My latent concern was that MarioNette was overly reliant on the grid of anchors carefully matching some ground truth tile size in order to recover good sprites. The library of extracted sprites for Mario seen in Figure 4 looked great: not too many apparent duplicates; boundaries that often felt correct; and more sprites allocated to objects that change appearance as they move (such as Mario) rather than items that simply scrolled (clouds and fences). By contrast, the sprites for the game shown in Figure 5 or Space Invaders at the bottom of Figure 6 felt significantly less clean and seemingly using multiple sprites to account for scrolling in a way that should have been handled as translations of a single cleaner sprite. In my head, this was explained by having implicitly fed the correct grid size for Mario and not for other games.
> > > >
> > > > Against this background it felt particularly suspicious that only half (1 of 2) of the datasets used to demonstrate SPACE were considered for comparative evaluation -- exactly the half where translation was the only transformation needed to re-form images with low reconstruction error (whereas 3d-room implicitly reuses objects with slightly different scales and lighting direction). Seeing MarioNette's behavior on 3d-rooms would help us understand the nature of the system.
> > > >
> > > > I'll revise my top-level review.

---

### Decision · Program_Chairs · 2021-09-27

**Decision:**

Accept (Poster)

**Comment:**

The submission proposes a self-supervised approach to represent images as a composition of elements from a dictionary of patches onto a canvas.

Reviewers felt that the paper tackles an interesting problem, is well-executed and provides convincing results for the video game animation domain. The main concern among reviewers is generalizability to more complex domains. The authors respond by pointing out that sprite-based animation and game content is already challenging enough for current approaches. They also point to Section 3.4 of the submission, which shows preliminary results on natural images and videos, noting that a richer image formation model would be required to obtain photorealistic results.

Reviewer vWXs was concerned that the proposed approach might be relying too much on task specificities to recover good sprites, which was addressed by the authors by clearing up confusion on how the data is scaled during preprocessing. Reviewer vWXs remains concerned that the translation-only design choice makes the approach brittle to variations in scale, pose, lighting, etc. between the training and evaluation settings, but notes that the approach can recover from that weakness through a better "transform predictor".

Given that overall reviewers agree that the submission advances the state-of-the-art with a new and interesting approach, I recommend acceptance.